# HIERARCHICAL LLM-GUIDED MULTI-TASK MANIPULATION WITH MULTIMODAL LEARNING AND ACTION-MASK POLICY

## ABSTRACT

Hierarchical policies that integrate high-level planning with low-level control have shown performance in robotic manipulation, but remain limited. We present a hierarchical framework that combines a two-stage task planner with a low-level action planner that integrates multimodal inputs and an explicit action-mask policy. At the high level, a Vision-Language Model (VLM) first perceives object and scene information from observations, and a Large Language Model (LLM) then reasons over this information together with a task library and human instructions to generate a textual task plan. This two-stage design mitigates modality bias. At the low level, we employ an asymmetric encoder, using SigLIP2 with Weight-Decomposed Low-Rank Adaptation (DoRA) for text and ResNets for multi-view vision. We introduce a shared Temperature-Scaled Spatial Attention module to enhance multi-view features and a Bidirectional Cross-Attention module to fuse language-vision features for Action Chunking Transformer (ACT) policy. For multi-task switching, we propose a novel explicit action-mask policy that jointly predicts actions and their validity masks. The policy learns not only fine-grained control but also when to stop, enabling real-time sub-task completion detection and robust switching across long-horizon tasks without additional inference overhead. Experiments on weighing and multi-object manipulation scenarios demonstrate planning accuracy, execution success, and efficiency, with ablations confirming the contribution of each component. Finally, deployment on a different robotic platform in a new scenario validates generalization. The video and code are available at `https://hierarchical-llm-robotics.github.io`.

## 1 INTRODUCTION

Completing complex tasks remains a challenge in robotic manipulation (Li et al., 2025; Jiang et al., 2025; Lee et al., 2021; Gu et al., 2025), as such settings often involve multiple objects, varying manipulation strategies, and long-horizon action sequences, etc. A promising framework to address these difficulties is the use of hierarchical policies (Kaelbling & Lozano-Pérez, 2011), where a high-level planner decomposes a complex task into a sequence of sub-tasks, and a low-level policy executes each sub-task in order. This approach requires a robust and capable high-level planner, as well as a multi-modal and precise low-level action generation policy, to ensure reliable performance.

Recent advances in large language models (LLMs) (Brown et al., 2020; Touvron et al., 2023; Team et al., 2023) have introduced new opportunities for robotic hierarchical planning. However, prior works (Zhou et al., 2025; Wang et al., 2025; Hu et al., 2023) leverage LLMs for task planning from observations and instructions, but often suffer from modality bias (Zhang et al., 2025b; Chen et al., 2024; Guo et al., 2023), over-relying on language reasoning while underutilizing visual information. Moreover, existing methods

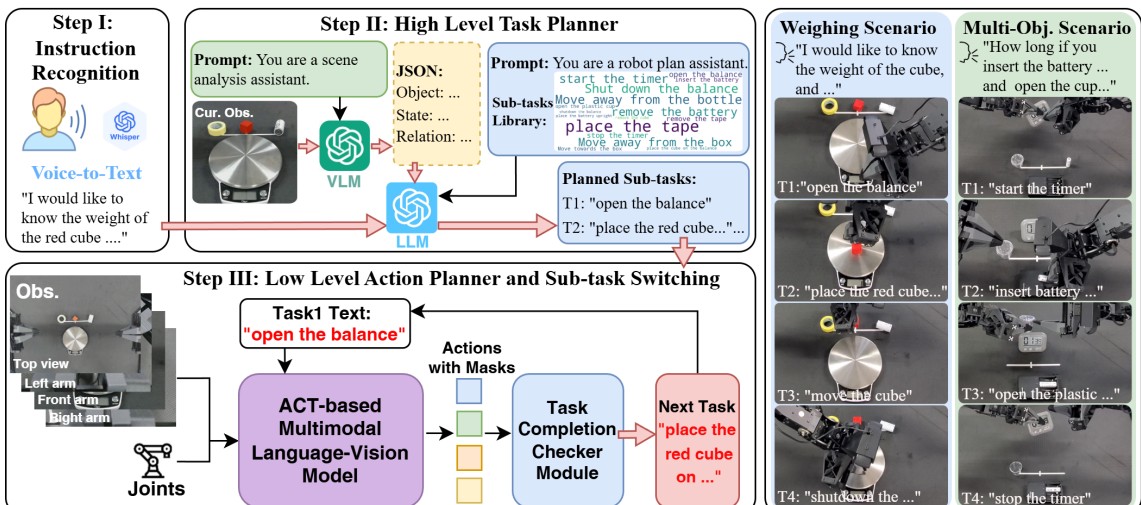

Figure 1: **Overview of the proposed hierarchical framework for language-guided, multimodal, multi-task robotic manipulation.** The framework operates in three process: (I) Instruction Recognition, where natural language commands are transcribed using a Whisper-based voice-to-text module; (II) High-Level Task Planning, where a VLM first perceives the scene to extract objects and information, and then an LLM, conditioned on the human instruction, the VLM output, and a library of predefined skills, decomposes the command into a sequence of sub-tasks; (III) Low-Level Action Generation and Sub-task Switching, where an ACT-based multimodal language-vision algorithm executes each sub-task by leveraging textual input, visual observations, and robot joint states. An explicit action-mask policy predicts both actions and their validity, while Task Completion Checker monitors these masks to detect termination and trigger switching to next sub-task. The right shows two examples of decomposed human instructions and their sub-task executions.

often rely on heuristic verification (Ha et al., 2023) or repeated LLM inference to switch sub-task (Zhang et al., 2025a; Guo et al., 2024; Shirai et al., 2024), which incurs computational overhead. Similarly, Vision-Language-Action (VLA) (Zitkovich et al., 2023; Kim et al., 2024) models have shown promise in aligning linguistic instructions with visual features, but many suffer from weak language-vision fusion (Shi et al., 2024; Ha et al., 2023) or high inference costs (Zhou et al., 2025), limiting their scalability in real-world applications.

To address this, we propose a hierarchical framework guided by a two-stage high-level planner and a low-level planner that combines language-vision multimodal learning with an explicit action-mask sub-task switching policy (see Figure1).

At the high level, to mitigate modality bias between vision perception and task planning, a VLM first performs visual inference to produce a structured scene description. The LLM then incorporates the human instruction and reasons over a sub-task library to generate a sequence of sub-task descriptions, which serve as input to the low-level policy.

At the low level, we design an asymmetric multimodal encoder that combines SigLIP2 (Tschannen et al., 2025) with DoRA (Liu et al., 2024) for text encoding and ResNet-based encoders for multi-view vision. A shared Temperature-Scaled Spatial Attention module employs spatial attention to enhance visual features. To fuse the textual and enhanced visual features, we propose a Bidirectional Cross-Attention module, which consists of two parallel cross-attention branches that enable each modality to attend to the other. This design

enables dynamic cross-modal interaction, enriching each modality with complementary information. The fused representation is then fed into the Action Chunking Transforme (ACT) (Zhao et al., 2023) policy.

For multi-task switching, we introduce a novel explicit action-mask policy, which jointly predicts both actions and their corresponding validity masks. By explicitly supervising the mask alongside action prediction, the policy learns not only how to generate fine-grained actions but also when an action sequence should terminate. This enables real-time sub-task completion detection, reducing overhead and improving efficient switching for long-horizon, multi-task manipulation.

We evaluate our framework on weighing and multi-object scenarios with diverse, long-horizon tasks, where it outperforms strong baselines in both planning accuracy, execution success, and efficiency. Ablation studies verify the contribution of each component, while deployment in another manipulation scenario on a different dual-arm platform demonstrates the generalization.

In summary, the main contributions of this study are:

1. A two-stage high-level planner where a VLM parses visual observation and an LLM reasons over instructions and a task library to generate structured task plans, mitigating modality bias between perception and planning.

2. An asymmetric multimodal encoder for the low-level planner, combining SigLIP2 with DoRA for text and multi-view ResNets for vision, together with shared Temperature-Scaled Spatial Attention and Bidirectional Cross-attention Models for language-vision fusion.

3. A novel explicit action-mask policy that jointly predicts actions and validity masks, enabling real-time sub-task completion detection without extra overhead and improving efficiency in multi-tasks switching.

4. Experiments confirm strong performance, ablations reveal each component's impact, while additional deployments on a different dual-arm robotic platform validate generalization.

## 2 RELATED WORK

### 2.1 HIERARCHICAL PLANNING AND SUB-TASK SWITCHING IN ROBOTICS

Hierarchical planning (Kaelbling & Lozano-Pérez, 2011) is an efficient approach for robotic manipulation in complex scenarios, as it decomposes tasks into a high-level planner and a low-level action controller. Recently, the integration of large language models (LLMs) (Duan et al., 2025; 2024; Zhou et al., 2024; Ahn et al., 2022) and advanced action generation models (Huang et al., 2024; Chi et al., 2023; Zhao et al., 2023) has further advanced this paradigm. Zhou et al. (2025); Wang et al. (2025); Hu et al. (2023) prompt LLMs with CoT reasoning, incorporating both observations and human instructions to generate task plans, demonstrating strong high-level planning ability. However, such approaches are prone to modality bias (Zhang et al., 2025b; Chen et al., 2024; Guo et al., 2023), where the model over-relies on prompted reasoning process while neglecting some visual information. For instance, a prompted LLM may detect objects in an observation image but fail to capture their states, such as whether a digital balance is powered on, whereas a dedicated perception stage can succeed (see Appendix A.3). To address this, we introduce a two-stage high-level method: a VLM first performs visual inference to describe the scene, and then an LLM reasons over the VLM output together with the human instruction and task library. This design mitigates modality bias by balancing perception and planning. Another challenge is sub-task switching, i.e., deciding when a sub-task is complete and transitioning to the next. Heuristic-based verification (Ha et al., 2023) often fails in unstructured settings, while relying on repeated LLM inference (Zhang et al., 2025a; Guo et al., 2024; Shirai et al., 2024; Schakkal et al., 2025) incurs high overhead. In contrast, we propose an explicit action-mask policy that predicts a validity mask for each action, enabling efficient sub-task completion detection and switching without extra computation.

## 2.2 VLAs for Robotics

Vision-Language-Action models (VLAs) (Pertsch et al., 2025; Din et al., 2025; O'Neill et al., 2024; Zitkovich et al., 2023) enable robots to learn multi-modal policies that associate language with visual observations, supporting open-vocabulary and multitask manipulation for improved robotic performance. A key challenge for VLAs lies in effectively aligning language with visual observations. For example, Ha et al. (2023) employ CLIP (Radford et al., 2021) to encode text directly, concatenating it with image features as input to a diffusion policy, while Shi et al. (2024) utilize FiLM (Perez et al., 2018) to fuse encoded visual features with DistilBERT (Sanh et al., 2019) language features. However, these methods do not finetune the text encoder, leading to weak alignment where visual features dominate the text. To address this, Zhou et al. (2025) leverage SigLIP (Tschannen et al., 2025), a model specifically designed for robust text-image alignment, by symmetrically employing it as the encoder for both text and images, together with DoRA (Liu et al., 2024) for joint finetuning across modalities. But their symmetrical encoder structure introduces significant parameter overhead, making real-time action generation, particularly with multi-view inputs, computationally impractical. In this work, we adopt an asymmetric encoder architecture: SigLIP2 with DoRA for text encoding and four ResNet-18 backbones for multi-view vision. We further introduce a shared spatial attention module and a bidirectional cross-attention module to strengthen cross-modal fusion, enabling robust performance in complex manipulation tasks.

## 3 Problem Statement

We aim to enable a robotic system to interpret natural-language instructions, plan a sequence of discrete sub-tasks from a predefined library, and execute them sequentially with automatic completion checking. Formally, given an instruction $\mathcal{L}_{\text{instr}}$ and a sub-task library $\mathcal{T}$, the high-level planner $\pi_H$ outputs

$$T_{\text{plan}} = (\tau_1, \ldots, \tau_m) = \pi_H(\mathcal{L}_{\text{instr}}), \quad \tau_i \in \mathcal{T}. \tag{1}$$

For each $\tau_i \in T_{\text{plan}}$, the low-level controller $\pi_L$ maps the observation $o_t = [o_{\tau_i}^{\text{text}}, o_{\tau_i,t}^{\text{cam}}, o_{\tau_i,t}^{\text{rob}}]$ to the next action

$$a_{t+1} = \pi_L(o_{\tau_i}^{\text{text}}, o_{\tau_i,t}^{\text{cam}}, o_{\tau_i,t}^{\text{rob}}). \tag{2}$$

A task-completion checker detects $n$ consecutive invalid actions and triggers transition to $\tau_{i+1}$ until all sub-tasks are finished.

## 4 Approach

### 4.1 High-Level Task Planner

We adopt a two-stage high-level planner, mitigates modality bias by separating perception from reasoning.

**Stage 1 (VLM visual perception).** The VLM is prompted as a scene-analysis assistant and receives the current observation, front-camera

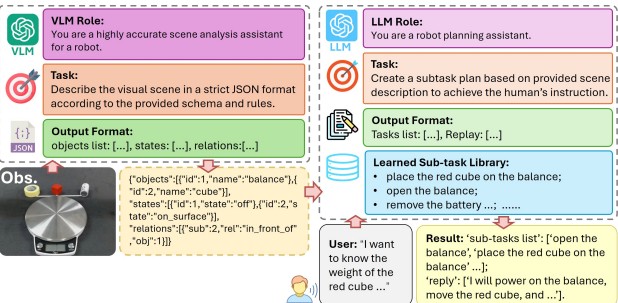

Figure 2: Two-stage high-level task planner.

image. It outputs a *strict JSON* description under a fixed schema, including `objects` (IDs, names), `states` (e.g., on/off, on-surface), and `relations` (e.g., in_front_of), as illustrated on the left of Figure 2.

**Stage 2 (LLM planning).** Conditioned on the JSON scene description, the human instruction obtained in Step I (Figure 1) via voice-to-text (Whisper), and a learned sub-task library $\mathcal{T}$, the LLM, prompted as

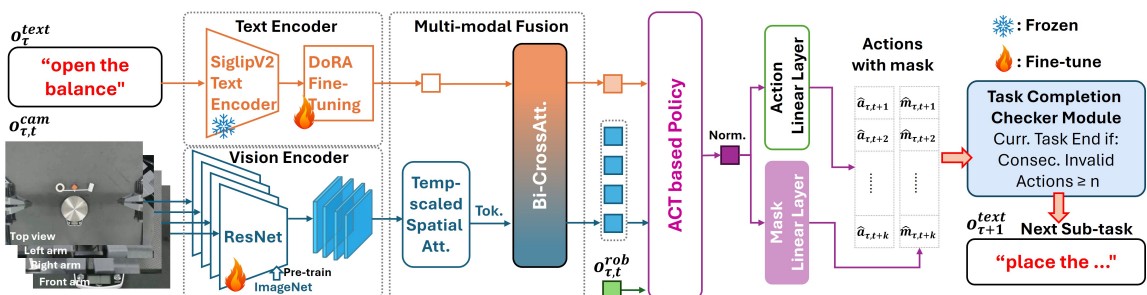

Figure 3: **Architecture of the Low-Level Action Planner.** For sub-task $\tau$ ("open the balance"), we adopt an asymmetric encoder: the language input is encoded by a frozen SigLIP2 (Tschannen et al., 2025) augmented with trainable DoRA adapters, while multi-view images at time $t$ is processed by a fine-tuned ResNet. The resulting features are fused via (i) a temperature-scaled spatial attention for visual enhancement and (ii) a bidirectional cross-attention for language-vision integration. These fused features, together with the robot joint state, are passed to an ACT-based policy (Zhao et al., 2023), whose output is normalized and augmented with two additional linear heads predicting the next $k$ timesteps of joint actions and their corresponding validity masks. A Task-Completion Checker monitors the masks and terminates the current sub-task if more than $n$ consecutive invalid actions are detected, switching to next sub-task $\tau + 1$.

a robot planning assistant, generates an ordered list of sub-tasks drawn from $\mathcal{T}$, together with a concise natural-language reply that explains the plan (see the right side of Figure 2).

This design balances perception and planning: the VLM provides structured visual facts, while the LLM reasons over these facts together with the instruction to produce executable, library-constrained plans. For example, given the instruction "I want to know the weight of the red cube ..." and a parsed scene where the balance is off and the cube is on-surface in back of it, the LLM outputs [open the balance, place the red cube on the balance, ...] and a short reply. For prompted details please see Appendix A.2.

### 4.2 LOW-LEVEL ACTION PLANNER

#### 4.2.1 ENCODER MODULE

We encode the sub-task text $\tau$ from the high-level planner with a SigLIP2-based text encoder (Tschannen et al., 2025). To adapt it to our policy, we fine-tune with DoRA (Liu et al., 2024), which decomposes weight updates into a low-rank term and a learnable magnitude, yielding greater expressivity with parameter efficiency. Concretely (Figure4a), for a linear projection with weight $W \in \mathbb{R}^{C \times K}$, we learn $A \in \mathbb{R}^{C \times r}$, $B \in \mathbb{R}^{r \times K}$, and a scaling vector $M \in \mathbb{R}^K$ (broadcast across rows), and compute $Y = X\big((W + AB) \odot M\big)$, where $X \in \mathbb{R}^{B \times C}$, $\odot$ denotes element-wise multiplication, $C$ is the input feature dimension, $K$ is the output feature dimension, and $r$ is the low rank. We insert DoRA adapters into the linear layers and fine-tune only $(A, B, M)$ while freezing the base SigLIP2 weights. For vision, we use ResNet-18 (He et al., 2016) backbones pretrained on ImageNet (Deng et al., 2009) and further fine-tuned on our task data. Finally, both text and vision features are projected to 512 dimensions for multimodal fusion later.

#### 4.2.2 MULTIMODAL FUSION MODULE

To obtain robust fused features for manipulation, we first enhance visual representations using a Temperature-Scaled Spatial Attention (TSA) module, and then integrate them with text via a Bidirectional Cross-Attention (BCA) module.

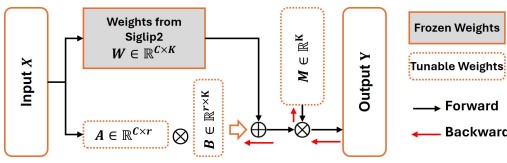

(a) SigLIP2 text encoder with DoRA adapters.

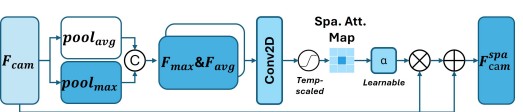

(b) Temperature-Scaled Spatial Attention.

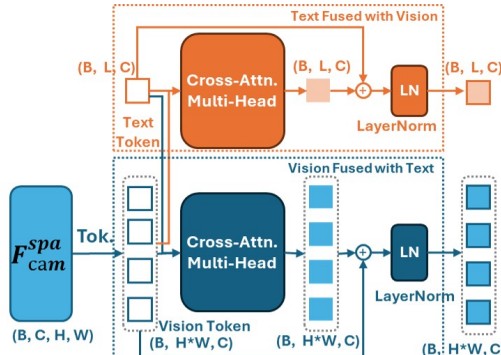

(c) Bidirectional Cross-Attention Module.

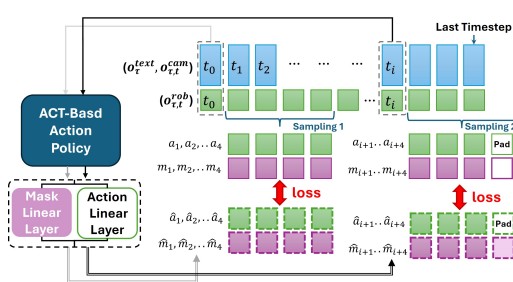

(d) Data sampling and loss computation.

Figure 4: Overview of key modules in low-level palnner

**Temperature-Scaled Spatial Attention (TSA)** employs a spatial attention mechanism to adaptively enhance feature representation. Given the encoded camera image features $F_{cam}$, two feature maps, $F_{cam}^{max}$ and $F_{cam}^{avg}$, are first obtained by applying maximum pooling and average pooling operations along the channel dimension, respectively. These two maps are concatenated and passed through a channel-reduction convolution to generate the attention logits. The logits are then scaled by a temperature parameter $\gamma$ and activated through $\tanh$ function to produce the temperature-scaled spatial attention map. Finally, the spatially attended feature $F_{cam}^{spa}$ is computed as $F_{cam}^{spa} = F_{cam} + \alpha \cdot \tanh\left(\frac{\text{attn\_logits}}{\gamma}\right) \odot F_{cam}$, where $\alpha$ is a learnable weight controlling the contribution of the attended features. In our setting, a single TSA module is shared across multi-camera views to enforce consistent attention and improve parameter efficiency.

**Bidirectional Cross-Attention (BCA)** enables mutual language-vision interaction via Transformer cross-attention (Vaswani et al., 2017; Lu et al., 2019), where each modality uses its features as queries to attend to the other. Figure4c shows the BCA architecture (matching the color-gradient box in Figure3). Two parallel branches perform cross-attention in both directions: the blue dotted box (with arrows) fuses visual features with textual ones, and the orange box denotes the reverse. Inputs are visual tokens from the TSA module and encoded text features.

### 4.2.3 EXPLICIT ACTION-MASK POLICY

The low-level planner not only generates continuous actions but also determines when to terminate the current sub-task and switch to the next. To this end, we introduce an explicit action-mask policy for real-time termination. For each predicted action, the policy outputs a binary mask $m \in \{0, 1\}$ indicating its validity, actions are gated by the mask, and the sub-task ends once a predefined number of consecutive invalid masks is observed.

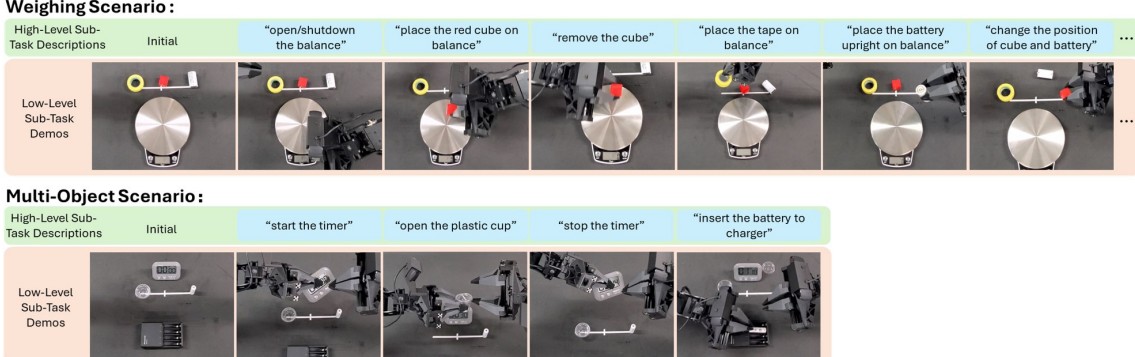

Figure 5: High-Level sub-task language descriptions and low-Level demonstrations for two scenarios.

Formally, the low-level policy $\pi_L$ maps the current observation tuple $o_{\tau_i}^{\text{text}}$, $o_{\tau_i,t}^{\text{cam}}$, $o_{\tau_i,t}^{\text{rob}}$ to the next action and its mask $(\hat{a}_{\tau_i,t+1}, \hat{m}_{\tau_i,t+1})$. We train $\pi_L$ with behavior cloning (BC). The objective is

$$\min_{\pi_L} \mathbb{E}_{(o_{\tau_i}^{\text{text}}, o_{\tau_i,t}^{\text{cam}}, o_{\tau_i,t}^{\text{rob}}, a_{\tau_i,t+1}, m_{\tau_i,t+1}) \sim \mathcal{D}} \left[ \mathcal{L}_{\text{BC}}\big( (\hat{a}_{\tau_i,t+1}, \hat{m}_{\tau_i,t+1}), (a_{\tau_i,t+1}, m_{\tau_i,t+1}) \big) \right]. \quad (3)$$

where $\mathcal{L}_{\text{BC}}$ combines an $\ell_1$ loss on actions and a binary cross-entropy-with-logits loss on the mask. This enables real-time termination without additional inference overhead. In this work, we use an ACT-based policy, which predicts the next $k$ action timesteps in a single forward pass. Therefore, the loss function is:

$$\mathcal{L}_{\text{BC}} = \text{mean}\big[ \big| \hat{a}_{\tau_i,t+1:t+k}, a_{\tau_i,t+1:t+k} \big| \big] + \lambda \, \text{mean}\big[ \ell_{\text{bce}}(\hat{m}_{\tau_i,t+1:t+k}, m_{\tau_i,t+1:t+k}) \big]. \quad (4)$$

The coefficient $\lambda$ weights the mask term, we use an annealing schedule that starts with a larger $\lambda$ and decays it during training.

Figure 4d shows the sampling and loss pipeline. We uniformly sample $k$-step (here $k=4$ for illustration) windows from demonstrations, padding incomplete segments with zeros and invalid masks. The ACT backbone outputs a shared representation, which we extend with two separate linear heads: one predicts actions (action linear layer) and the other predicts mask probabilities (mask linear layer). We optimize with an $\ell_1$ loss on actions and BCE on masks, enabling the policy to jointly learn control and termination. Further details are provided in Appendix A.4. At inference, sub-task completion is detected once the mask signals $n$ consecutive invalid actions, triggering the Task Completion Checker to switch sub-tasks (Figure 3).

## 5 EXPERIMENTS AND RESULTS

### 5.1 TASKS DESCRIPTION

We design two manipulation scenarios for our method on our dual-arm robotic platform (Appendix A.5).

**Weighing Scenario.** Weighing objects (tape, cube, battery) on a digital balance, which may be powered off, requires sub-tasks such as powering the balance and placing/removing objects (Figure 5, top). This setting tests high-level decomposition and low-level generalization across diverse manipulation strategies.

**Multi-Object Scenario.** Tasks include operating a timer, opening a cup, and installing a battery (Figure 5, bottom). Given instructions (e.g., "Tell me how long if you install the battery"), the planner must sequence actions such as starting the timer, inserting the battery, and stopping it, stressing long-horizon, fine-grained execution.

| Method | Weighing Scenario | | | | | | | Multi-Object Scenario | | | | | | |
|---|---|---|---|---|---|---|---|---|---|---|---|---|---|---|
| | High-Level (Succ. %↑) | | | Time [s]↓ | Low-Level (Succ. %↑ / Time [s]↓) | | | High-Level (Succ. %↑) | | | Time [s]↓ | Low-Level (Succ. %↑ / Time [s]↓) | | |
| | T1 | T2 | T3 | | T1 | T2 | T3 | T1 | T2 | T3 | | T1 | T2 | T3 |
| Flat-ACT | – | – | – | – | 0 / – | 0 / – | 0 / – | – | – | – | – | 0 / – | 0 / – | 0 / – |
| SayCan | 80 | 70 | 70 | 15.5 | – | – | – | 80 | 75 | 70 | 14.9 | – | – | – |
| One-stage CoT | 80 | 75 | 70 | **7.8** | – | – | – | 85 | 70 | 70 | **8.3** | – | – | – |
| YAY | – | – | – | – | 80 / 87.2 | 75 / 128.9 | 70 / 166.5 | – | – | – | – | 75 / 128.7 | 70 / 183.8 | 60 / 259.1 |
| Ours | **100** | **95** | **85** | 13.5 | **95 / 62.7** | **95 / 99.7** | **90 / 126.8** | **100** | **100** | **90** | 12.7 | **90 / 86.4** | **85 / 132.5** | **80 / 191.1** |

Table 1: Comparison results. "–" indicates that the method does not support the corresponding evaluation. T1, T2, and T3 denote Tier 1, Tier 2, and Tier 3, respectively.

## 5.2 COMPARATIVE EVALUATION

**Metrics:** We evaluate performance using three metrics: (i) high-level plan success rate, (ii) low-level execution success rate (assuming a correct plan), and (iii) average inference/execution time. Precise definitions are provided in Appendix A.7.

**Evaluation Tiers:** Instructions are grouped into three tiers (Tier 1, Tier 2, Tier 3) of increasing complexity (2-4 sub-tasks). More details are provided in Appendix A.8.

**Comparison Methods:** We compare against four baselines: (i) Flat-ACT (Zhao et al., 2023), a flat policy trained on combined demonstrations; (ii) SayCan (Ahn et al., 2022), value-based LLM planning; (iii) One-Stage CoT, where an LLM directly plans via CoT; and (iv) YAY (Shi et al., 2024), used for low-level comparison. More details are in Appendix A.9.

**Results:** Table 1 reports results across three tiers under two scenarios, with 20 trials per setting. Flat-ACT highlights the limitations of flat policies: without hierarchical decomposition, it fails to generalize in multi-task settings. SayCan, which relies on a value-based verification method, and One-stage CoT both underperform compared to our approach, although One-stage CoT achieves the shortest inference time. In contrast, our two-stage high-level planner consistently delivers the highest success rates across all instruction tiers, albeit with longer inference time. For low-level execution, YAY is a strong baseline but suffers from lower success rates and longer execution times, as it lacks an termination mechanism and must wait until the timestep end of each sub-task. Our low-level planner outperforms YAY in both precision and efficiency across two manipulation scenarios. Moreover, the explicit action-mask policy improves efficiency and reliability by predicting termination, which reduces redundant motions and shortens execution time.

## 5.3 ABLATION

We conduct four ablations. Results are summarized in Table 2, with details in Appendix A.10. **LLMs for High-Level Planner:** We compare Qwen, Gemini, GPT-5, and GPT-4o. GPT series yields the highest success, with GPT-5 slightly stronger but much slower. We adopt GPT-4o as a balanced choice. **Encoders for Text and Vision:** Our asymmetric design (SigLIP2&DoRA for text, ResNets for vision) outperforms all methods in success rate, while the symmetric SigLIP2&DoRA variant is impractical due to the parameter increase (120.24M → 449.95M), resulting significant inference time. **Multimodal Fusion:** Removing TSA or BCA reduces success rate, showing both modules are essential for accurate manipulation and robust fusion. **Sub-task Switching:** Replacing our action mask with LLM-based checks or fixed timesteps leads to longer execution times. Our mask-based approach enables efficient and reliable sub-task switching.

| Method | High-Level (Succ. %↑) | | | Time [s]↓ |
|---|---|---|---|---|
| | T1 | T2 | T3 | |
| **GPT-5** | **100** | **100** | **90** | 73.0 |
| **GPT-4o** | **100** | 95 | 85 | 13.5 |
| **Qwen** | **100** | 70 | 40 | 35.7 |
| **Gemini** | 90 | 90 | 80 | **6.5** |

(a) LLMs for High-Level Planner (High-level)

| Method | Low-Level (Succ. %↑ / Time [s]↓) | | |
|---|---|---|---|
| | T1 | T2 | T3 |
| **Ours (Asymmetric)** | **95** / 62.7 | **95** / 99.7 | **90** / 126.8 |
| **SigLIP2&DoRA (Symmetric)** | 0 / - | 0 / - | 0 / - |
| **SigLIP2 (Text)** | 90 / 61.1 | 85 / 99.8 | 80 / 124.5 |
| **DistilBERT (Text)** | 85 / **58.8** | 85 / **95.7** | 75 / **117.0** |

(b) Encoders for Text and Vision (Low-level)

| Method | Low-Level (Succ. %↑ / Time [s]↓) | | |
|---|---|---|---|
| | T1 | T2 | T3 |
| **Ours** | **95** / 62.7 | **95** / 99.7 | **90** / 126.8 |
| **No TSA** | 95 / 63.2 | 90 / 100.2 | 80 / 124.2 |
| **No BCA** | 90 / **62.1** | 85 / 100.8 | 75 / **119.5** |

(c) Multimodal Fusion (Low-level)

| Method | Low-Level (Succ. %↑ / Time [s]↓) | | |
|---|---|---|---|
| | T1 | T2 | T3 |
| **Ours** | **95** / 62.7 | **95** / 99.7 | **90** / 126.8 |
| **LLM-based** | **95** / 86.9 | **95**/ 133.4 | 85 / 167.8 |
| **Fixed-ts** | **95** / 88.2 | 90 / 132.2 | **90** / 170.5 |

(d) Sub-task Switching (Low-level)

Table 2: Ablation studies under the weighing scenario.

## 5.4 GENERALIZATION TO OTHER PLATFORM AND EXPERIMENT

We further validate our approach on a different platform in a drawer-storage scenario (Figure 6). Additional details are provided in the Appendix A.11 and video.

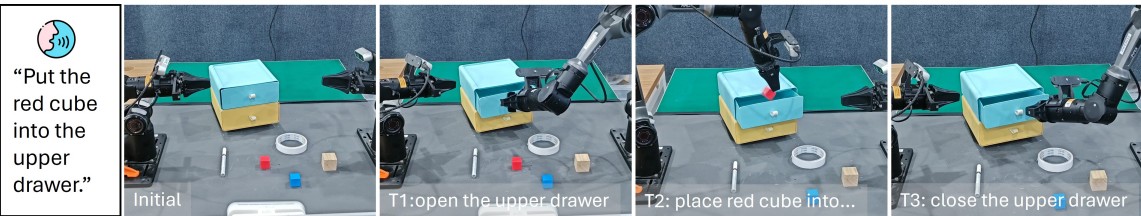

Figure 6: Deployment in a drawer-storage scenario using another robotic platform.

## 6 CONCLUSION

Hierarchical framework with high-level task planner and low-level action planner can enhance complex multi-task robotic manipulation. We introduce a two-stage high-level task planner combines with a Vision-Language Model (VLM) and Large Language Model (LLM) , could mitigate modality bias. On the low-level side, an asymmetric encoder design, SigLIP2 with Weight-Decomposed Low-Rank Adaptation (DoRA) for text and multi-view ResNets for vision, cuts compute while preserving strong language-vision alignment. A Temperature-Scaled Spatial Attention module to enhance multi-view features and a Bidirectional Cross-Attention module to fuse language-vision features. A novel explicit action-mask policy for Action Chunking Transformer (ACT) jointly learns continuous actions and validity masks, enabling real-time sub-task completion detection and robust switching without additional inference overhead. Experiments confirm advantages over strong baselines, with ablations validating each module and deployment on another dual-arm platform demonstrating generalization.

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

## A APPENDIX

### A.1 THE USE OF LARGE LANGUAGE MODELS

In this work, Large Language Models (LLMs) are not only studied for high-level task planning within our framework but are also employed for checking grammar and polishing the writing of this manuscript.

### A.2 PROMPTS FOR OUR TWO-STAGE HIGH-LEVEL PLANNER AND THE ONE-STAGE CoT BASELINE

We provide the exact prompts used in our experiments for reproducibility. Our proposed two-stage planner separates perception from reasoning, while the one-stage baseline directly performs planning with chain-of-thought (CoT) reasoning.

**Our Proposed Two-Stage Prompt:**

Stage 1: Perception-only prompt.

```
system_prompt = (
    "You are a highly accurate scene analysis assistant for a robot. "
    "Your task is to describe the visual scene in strict JSON format. "
    "Only describe what you see. Do not infer intent or plan tasks."
)

user_text = (
    "Please analyze the provided image and describe it using the following
        schema:\n"
    "{ 'objects': [...], 'states': [...], 'relations': [...] }\n"
```

```
"Constraints:\n"
"1. IDs start from 1.\n"
"2. States must follow predefined values (on/off, on_surface, etc.).\n"
"3. Relations use spatial prepositions like 'on_top_of' or 'next_to'.\n"
)
```

Stage 2: Planning-only prompt.

```
system_prompt = (
    "You are a robot planning assistant. "
    "Your task is to create a plan from a provided scene description "
    "to achieve a human's instruction. "
    "Only use the known instructions."
)

user_text = (
    f"Human instruction: {user_instruction}\n\n"
    f"Scene description: {perception_str}\n\n"
    f"Known instructions: {instruction_listing}\n\n"
    "Output JSON with 'tasks' and 'reply'."
)
```

**The Baseline of One-Stage Prompt with CoT:**

```
system_prompt = (
    "You are a robot planning assistant with visual perception. "
    "First, internally perceive the scene from provided images (do not reveal
        your chain-of-thought). "
    "Second, plan a sequence of known robotic instructions to achieve the human
        instruction. "
    "Only output a single JSON object matching the required schema. "
    "Never include your reasoning process or any explanation outside the JSON."
)

user_text = (
    "Task:\n"
    f"- Human instruction: {user_instruction}\n\n"
    "Known instructions (choose zero or more, order matters):\n"
    f"{instruction_listing}\n\n"
    "Pipeline (use implicit reasoning, do NOT reveal the reasoning):\n"
    "1) Perception from images (internal-only):\n"
    "   a) What objects are in the scene?\n"
    "   b) What are their states (on/off, open/closed, etc.)?\n"
    "   c) What are the spatial relations?\n"
    "2) Planning: choose and order the known instructions.\n\n"
    "Output format (JSON ONLY): {...}\n"
)
```

A.3    COMPARISON OF TWO-STAGE HIGH-LEVEL PLANNER WITH ONE-STAGE METHOD

As illustrated in Figure 7, the one-stage LLM-based method with CoT reasoning fails to perceive the state of the digital balance, resulting in an incorrect plan that redundantly includes the sub-task "turn on the balance." This error arises from modality bias, where the LLM focuses on abstract reasoning while neglecting

visual perception, particularly when the visual information is ambiguous. In contrast, our two-stage planner explicitly separates perception and reasoning: the VLM first infers that the balance is already "on," and the LLM then generates a correct task plan without unnecessary actions.

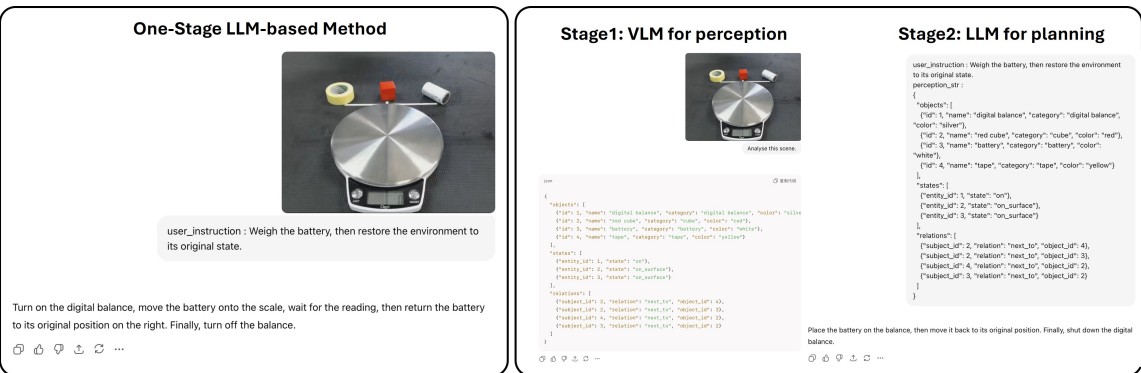

Figure 7: Comparison of planning results between the standard one-stage LLM-based CoT method and our proposed two-stage planner. The one-stage LLM-based CoT method fails to observe the correct power state of the balance, whereas our two-stage planner accurately perceives it and generates a correct task plan. For clarity, the results are obtained using the web-based GPT interface.

### A.4 DETAILS FOR ACTION-MASK LOSS COMPUTATION AND INFERENCE

Figure4d illustrates the data sampling and loss computation pipeline. We uniformly sample windows from demonstrations: blue blocks denote the observation tuple (camera images and instruction text), and green blocks denote robot joint states. At time $t_0$, the ACT-based policy predicts a $k$-step action chunk (here $k=4$ for illustration). The next four joint states serve as action targets (purple). Feeding the observation at $t_0$ into the low-level planner yields predicted actions and masks (green/purple dashed boxes), and we compute the loss against the ground-truth targets. In the second case, we sample a window at a later time $t_i$, closer to the end of the demonstration. Since only three future steps remain, we pad the fourth action target with zeros and mark its corresponding mask as invalid (purple outlined box), indicating task completion is near.

During training, the action loss is averaged over the available (non-padded) steps, while the mask loss is computed over all $k$ steps using binary cross-entropy with logits. Concretely, the ACT backbone outputs a shared normalize representation, which we extend with two separate linear heads: one predicts actions (action linear layer) and the other predicts mask probabilities (mask linear layer). This setup allows the model to jointly learn both fine-grained action prediction and when to terminate the task, guided by the mask signal.

During inference, the policy outputs a chunk of predicted actions along with their corresponding mask probabilities indicating action validity. If the mask probability exceeds a predefined threshold and the number of consecutive invalid actions reaches $n$, the sub-task is considered complete. Control then transitions to the next sub-task, as handled by the Task Completion Checker Module (see Figure3).

### A.5 ROBOTIC SYSTEM SETUP

We built our system based on the Aloha (Zhao et al., 2023). Unlike the original Aloha system, where the top-view camera is mounted on the metal frame, we mount it on a fixed boom (see right image), which helps avoid visual vibrations caused by robot movement. We further add a microphone to capture human language

Figure 8: Description of our robotic platform.

instructions. For vision, four Logitech C922n webcams are used: two wrist-mounted for close-up object views, and two front/top cameras for scene coverage. The system operates at 50 Hz, with a monitor for operator feedback and a foot pedal that allows the operator to manually end recording once a demonstration is complete. Demonstrations are recorded via a bimanual teleoperation setup with two master and two slave arms, enabling natural control while logging synchronized text, vision, and joint data for imitation learning.

## A.6 DETAILS OF DATASET, TRAINING, AND DEPLOYMENT

We collect demonstrations using our robotic system. For both weighing and multi-object scenarios, each sub-task is recorded separately, with 40 demonstrations per sub-task (50 for more complex ones such as inserting a battery). Episodes end when the operator completes the task and presses the foot pedal.

We train the policy on a server with RTX A6000 GPUs. The model is trained using Adam with a learning rate of $1.0 \times 10^{-5}$, batch size 32, and action chunk size 100 for ACT, for 100k iterations.

During deployment, we use a workstation with an RTX 3090 GPU and 64GB RAM. For temporal smoothing, we adopt ACT's (Zhao et al., 2023) temporal ensembling with a reduced weight (default $1.0e-2$) to emphasize recent predictions, improving reactivity at the cost of slight smoothness. For task termination, we set the invalid mask probability threshold to 0.75. If three consecutive actions are marked invalid, the system switches to the next task.

## A.7 DETAILS FOR METRICS

We evaluate our system using three metrics:

- **High-Level Plan Success Rate**: Measures whether the high-level planner outputs the correct sequence of sub-tasks for a given instruction, as judged by human.

- **Low-Level Execution Success Rate**: Measures the ability of the low-level policy to successfully complete the sub-tasks, assuming a correct task plan is provided.
- **Average Inference and Execution Time**: Reports the average inference time of the high-level planner and the average execution time of the low-level policy, computed only over attempts in which all sub-tasks are completed.

### A.8 DETAILS FOR EVALUATION TIERS

We evaluate performance across three tiers of instruction complexity:

- **Tier 1:** Instructions require **2 sub-tasks** to complete. *Example:* "Please help me restart the balance," which involves shutting down and turning it back on.
- **Tier 2:** Instructions require **3 sub-tasks**. *Example:* "Please tell me the weight of the cube," assuming the balance is already powered on. The low-level planner handles placing the cube, waiting for measurement, and returning it.
- **Tier 3:** Instructions require **4 sub-tasks**. *Example:* "Please tell me the weight of the No.2 battery, and remember to put everything back afterward," which includes turning on the balance, placing the battery upright, removing it, and turning off the balance.

### A.9 DETAILS FOR COMPARSION METHODS:

We compare our method against three baselines:

- **Flat-ACT:** To assess the benefit of hierarchical planning, we compare against a flat policy based on Action Chunking with Transformers (ACT) (Zhao et al., 2023), a SOTA imitation learning method. We train ACT on combined sub-task demonstrations to execute full instructions without hierarchical decomposition.
- **SayCan:** To evaluate high-level planning, we include SayCan (Ahn et al., 2022), which uses LLMs for sub-task selection via language-based reasoning combined with a value function conditioned on visual and text inputs. We only compare the high-level planning component, as the low-level policy is not the focus of their work.
- **One-Stage CoT:** A widely used baseline where the LLM directly generates sub-task sequences from raw observations and instructions via Chain-of-Thought (CoT) reasoning, without a separate perception stage.
- **YAY:** To compare low-level capabilities, we adopt the low-level policy of YAY (Shi et al., 2024), a strong vision-language-based algorithm. We use the same sub-task instructions for fair comparison.

### A.10 DETAILED ABLATION STUDIES

We conduct four ablations to validate the design choices of our framework. Here we provide the complete settings and results for each study.

**LLMs for High-Level Planner:** To evaluate the effectiveness of different LLMs as high-level task planners, we benchmark four recent models: Qwen (Hui et al., 2024), Gemini (Comanici et al., 2025), GPT-5 (OpenAI, 2025), and GPT-4o (Hurst et al., 2024). For each model, we employ two-stage inference and conduct 20 trials per tier in the weighing scenario. As shown in Table 2a, the GPT series achieves the highest success rates, but GPT-5 incurs the longest inference time. Considering this trade-off, we adopt GPT-4o as the high-level task planner.

**Encoders for Text and Vision:** We vary the encoder of the low-level planner using a symmetric structure, where both text and vision are encoded with SigLIP2 fine-tuned with DoRA. We also test variants that

replace SigLIP2&DoRA with plain SigLIP2 or DistilBERT for text. As shown in Table 2b, our method achieves the highest success rate, while the symmetric SigLIP2&DoRA for text and vision design is impractical due to the parameter increase (120.24M → 449.95M) and resulting very high inference time.

**Multi-modal Fusion Strategy:** We ablate the proposed Temperature-Scaled Spatial Attention (TSA) and Bidirectional Cross-Attention (BCA) modules individually by removing them from the architecture. We ablate TSA and BCA separately, removing one module at a time while keeping the other unchanged The ablation results are reported in Table 2c. The result shows that the absent of TSA and BCA lead to decrease success rate, because of the precise decrease and multimodal fusion inference., while maintain the similar execution time.

**Action Mask for Sub-task Switching:** We replace the action mask with two strategies: (1) an iterative LLM-based completion check, where a lightweight Gemini-2.0-Flash-Lite model (Team et al., 2025) verifies sub-task completion at 1 Hz (Schakkal et al., 2025), and (2) using the default pre-defined fixed timestep lengths for each sub-task (Fixed-ts). As shown in Table 2d, both alternatives incur longer execution times, whereas our explicit action-mask mechanism enables immediate termination and switching upon sub-task completion, leading to higher efficiency.

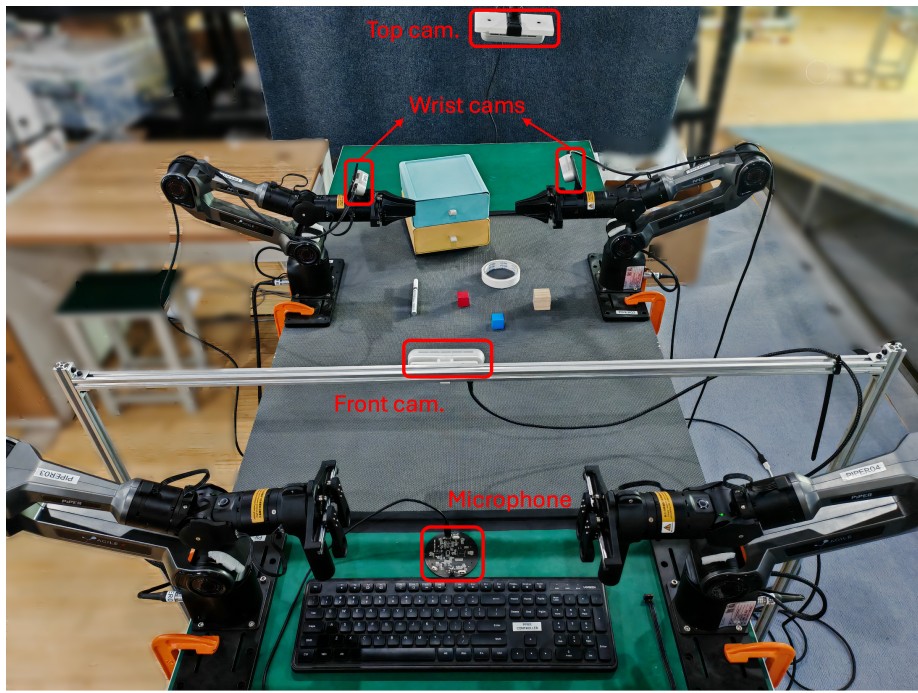

Figure 9: Robotic platform for generalization

## A.11 DETAILS FOR GENERALIZATION TO OTHER PLATFORM AND EXPERIMENT

**Another Robotic System Setup:** For the robotic platform used in generalization, please refer to Figure 9. We employ a PIPER 6-DoF robotic arm from AgileX, which supports CAN bus communication between paired arms to enable ALOHA-style master, slave teleoperation for data collection. For perception, the

system incorporates four Intel RealSense cameras: D435i models mounted on the wrists and D455f models positioned for the top and front views.

**Task Description:   Storage Scenario.** The task involves organizing scattered items on a desktop (e.g., pens, blocks, tape) into designated drawers as required. This scenario can be decomposed into two main sub-task types: opening/closing specific drawers and picking/placing designated objects. Unlike standard pick-and-place tasks, this setup challenges the algorithm's ability in high-level semantic understanding, sequential planning, and generalization across diverse objects.

