# OpenReview forum: "Hierarchical LLM-Guided Multi-Task Manipulation with Multimodal Learning and Action-Mask Policy"
_ICLR.cc/2026/Conference — ICLR 2026 Conference Withdrawn Submission_

### Official Review · Reviewer_2UE3 · 2025-10-28

**Soundness:** 3
**Presentation:** 3
**Contribution:** 2
**Rating:** 4
**Confidence:** 4

**Summary:**

This work presents a comprehensive framework for addressing long-horizon and complex tasks, consisting of two main components: a high-level planner and a low-level policy, with an additional voice control module.
The high-level planner is responsible for scene understanding and task decomposition, while the low-level policy introduces a language-conditioned, act-based model for action execution.

**Strengths:**

1. The overall structure of the paper is well-organized, and the logic is clear.

2. The method is explained clearly, allowing readers to quickly understand the detailed approach of the work.

3. This is a complete and systematic piece of research.

**Weaknesses:**

1. The proposed method feels somewhat “straightforward.” Equipping a VLA model with a high-level planner is essentially an internal mechanism for handling long-horizon tasks, which makes it difficult to view as a core innovation. Moreover, the feature fusion design in the low-level policy also appears to have limited novelty.

2. The experimental setup is not clearly described — for example, it is unclear what fine-tuning data were used, how large the dataset is, and other relevant details.

**Questions:**

1. I find the action-mask policy used as a task completion detector somewhat confusing. Why does adding a mask enable the model to determine whether a task has been completed? Shouldn’t task completion be judged based on visual input or certain state information instead?

2. Fine-tuning a pretrained encoder (such as the SigLIP2 text encoder) is not a common practice. Could you explain why fine-tuning is necessary in this case?

3. The experimental setup is not clearly described. I am unsure what fine-tuning data were used, how large the dataset is, and what the detailed training configurations are.

Lastly, a minor suggestion: this work feels more suitable for ICRA or IROS, as it represents a complete system-level effort but lacks an element of novelty or conceptual intrigue.

---

### Official Review · Reviewer_XESx · 2025-10-29

**Soundness:** 3
**Presentation:** 3
**Contribution:** 3
**Rating:** 4
**Confidence:** 3

**Summary:**

This paper proposes a hierarchical robotic manipulation framework that integrates large language models (LLMs) and vision-language models (VLMs) for multi-task, long-horizon manipulation. The method separates perception (via a VLM) and reasoning/planning (via an LLM) to mitigate modality bias. At the low level, the framework introduces (1) an asymmetric multimodal encoder (SigLIP2 + DoRA for text and ResNet-based vision), (2) a Temperature-Scaled Spatial Attention (TSA) and Bidirectional Cross-Attention (BCA) for fusion, and (3) an explicit action-mask policy that jointly predicts actions and termination masks for sub-task switching. Experiments on dual-arm robot setups (weighing and multi-object manipulation) show improvements in task success and efficiency, with ablations validating the contribution of each component and generalization shown on a second platform.

**Strengths:**

- The separation between VLM perception and LLM planning is well-justified to combat modality bias, which is an important issue in VLM–LLM integration for robotics.
  - The two-stage planning pipeline (VLM for structured scene representation → LLM for task reasoning) is elegant and interpretable.
2. Novel low-level policy design.
  - The explicit action-mask mechanism for real-time sub-task completion is a practical and innovative contribution that directly addresses inefficiencies in long-horizon control.
  - The asymmetric encoder architecture balances computational cost and multimodal expressivity, which is valuable for real-world deployment.
3. Comprehensive experiments.
  - Includes comparisons to strong baselines (SayCan, YAY, CoT planning, ACT).
  - Ablation studies are systematic and isolate contributions of LLM choice, encoder symmetry, TSA/BCA fusion, and mask policy.
  - The cross-platform validation adds credibility to the generalization claim.
4. Strong engineering execution and clarity.
  - Figures and diagrams (Figures 1–4) are informative and easy to follow.
  - Prompts are provided in the appendix, supporting reproducibility.

**Weaknesses:**

1. Limited novelty at the conceptual level.
  - While the paper’s combination of known ideas (hierarchical LLM-VLM structure, cross-attention fusion, mask-based termination) is effective, the individual components are mostly incremental extensions of existing work (e.g., SayCan, ACT, SigLIP-based fusion).
  - The explicit action-mask policy resembles termination gating or validity prediction seen in prior action-chunking or skill-switching literature (e.g., hierarchical RL termination functions).
2. Evaluation scope is relatively narrow.
  - Only two primary scenarios (weighing and multi-object) are tested, both tabletop and dual-arm setups with limited object diversity.
  - Tasks involve mainly short sequences (≤4 sub-tasks), so claims about “long-horizon manipulation” may be overstated.
3. Insufficient quantitative analysis of planning vs. control contributions.
  - While ablations isolate modules, it remains unclear how much of the performance improvement comes from high-level planning vs. low-level policy.
  - The success metrics mix perception, reasoning, and control success; more fine-grained metrics (e.g., sub-task detection accuracy, action validity prediction F1) would strengthen the analysis.
4. No real comparison to existing hierarchical frameworks.
  - Missing comparisons to HAMSTER (ICLR 2025) or Robridge (arXiv 2025a), which also use hierarchical multimodal reasoning for manipulation.
  - These would provide stronger evidence that the proposed hierarchical integration offers unique benefits.
5. Writing and presentation.
  - While overall clear, some sections (especially in Appendix A.10) contain minor grammatical errors and redundancy.
  - The “Conclusion” section reads more like a restatement of contributions than a discussion of implications or limitations.

**Questions:**

1. How does the action-mask policy compare to learned termination functions in hierarchical reinforcement learning (e.g., options framework)?

2. How robust is the system to VLM misperception? Does an incorrect JSON scene description propagate to major planning errors?

3. What is the real-time inference latency for both planners combined? Can this system operate interactively in real-world settings (>5 Hz)?

4. Did you attempt fine-tuning the LLM on task planning examples? Or is it purely prompt-based zero-shot reasoning?

5. Could the method extend to continuous high-level planning (without discrete sub-task libraries)?

---

### Official Review · Reviewer_GEYv · 2025-10-30

**Soundness:** 3
**Presentation:** 3
**Contribution:** 2
**Rating:** 2
**Confidence:** 5

**Summary:**

In this paper, the authors proposed a pipeline consisting of a high-level task planner with a low-level action planner to do manipulation tasks. To be specific, the authors used a VLM to extract visual features and an LLM to decompose the task for the low-level action head. Then, the authors trained a low-level action head based on the ACT, modifying the vanilla ACT with multi-modal inputs. The authors perform real-world experiments in weighing scenario and multi-object scenario, demonstrating the effectiveness of the proposed method.

**Strengths:**

1. The paper is well-written and easy to follow.
2. Compared to the baselines, the proposed method improves performance effectively.

**Weaknesses:**

1. The method's novelty is limited. The idea of combining high-level planner with low-level action head is broadly explored in the previous methods. Also, the multi-modal ACT conditioned on the text inputs is also common and not novel.
2. The scope of experiments is limited. The experimental scenarios are relatively simple, which cannot be considered as challenging to the VLM and LLM.
3. The baselines are missing. I noticed the authors did not compare with concurrent VLA methods like pi0, pi0.5 or Gr00T etc..

**Questions:**

See the weakness.

---

### Official Review · Reviewer_YBDt · 2025-10-30

**Soundness:** 2
**Presentation:** 2
**Contribution:** 2
**Rating:** 4
**Confidence:** 3

**Summary:**

This paper introduces a hierarchical framework for long‑horizon robot manipulation that separates perception from reasoning: a VLM first converts the scene into a structured JSON description, and an LLM turns that into an ordered sequence of sub‑tasks from a predefined library. Execution is handled by an asymmetric multimodal policy that encodes language (SigLIP2+DoRA) and multi‑view vision (ResNets), fuses them with temperature‑scaled spatial attention and bidirectional cross‑attention, and adds an action‑mask head that predicts when actions are invalid so the system can stop a sub‑task and switch without extra inference. Across multi‑step tabletop tasks (weighing objects and multi‑object manipulation), the approach achieves higher planning accuracy and faster, more reliable execution than baselines, with ablations showing the two‑stage planner, fusion modules, and action‑mask are key contributors and a deployment on a different robot indicating some generalization.

**Strengths:**

It cleanly separates perception from reasoning (VLM → LLM), which reduces modality bias and yields more reliable, library-constrained plans; it introduces an action-mask that lets the policy detect sub-task completion and switch tasks without extra LLM calls, making execution faster and more stable; its asymmetric encoders plus TSA/BCA fusion tighten language-vision alignment; results show higher planning success and shorter execution times than strong baselines across multi-step tabletop tasks; and ablations plus a cross-robot demo support both the design choices and some generalization.

**Weaknesses:**

- Relies on a fixed sub-task library, limiting open-ended generalization to unseen skills.
- Requires per–sub-task teleop demonstrations and a multi-stage pipeline, adding data and engineering overhead.
- Sub-task switching depends on an action‑mask threshold (e.g., n consecutive invalids), which can be sensitive to noise and may need tuning.
- Two‑stage planning improves accuracy but adds latency; also introduces dependence on large, possibly closed and costly LLMs.
- Evaluation is confined to a few tabletop scenarios with modest trial counts; evidence beyond a single cross‑platform demo is limited.
- Robustness/safety under disturbances, sensor noise, or environment shift is not extensively analyzed.

**Questions:**

See weakness

---

### Note · Authors · 2025-11-13

I have read and agree with the venue's withdrawal policy on behalf of myself and my co-authors.